Methods

# Screening of homing and tissue-penetrating peptides by microdialysis and in vivo phage display

Toini Pemmari[1,2,*] , Stuart Prince[1,2,*] , Niklas Wiss[1,2,*] , Kuldar Kõiv[3], Ulrike May[1,2], Tarmo Mölder[3], Aleksander Sudakov[3], Fernanda Munoz Caro[1,2] , Soili Lehtonen[1,2], Hannele Uusitalo-Järvinen[1,2] , Tambet Teesalu[3], Tero AH Järvinen[1,2]

In vivo phage display is a method used for identification of organ- or disease-specific vascular homing peptides for targeted delivery of pharmaceutics. It is agnostic as to the nature and identity of the target molecules. The current in vivo biopanning lacks inbuilt mechanisms to select for peptides capable of vascular homing that would also be capable of tissue penetration to reach therapeutically relevant cells in the tissue parenchyma. Here, we combined in vivo phage display with microdialysis-based parenchymal recovery and high-throughput sequencing to select for peptides that, besides vascular homing, facilitate extravasation and tissue penetration. We first demonstrated in skin wounds that the method can selectively separate known homing peptides from those with additional tissue-penetrating ability. Screening of a naïve peptide library identifies peptides that home and extravasate to extravascular granulation tissue in vascularized and diabetic wounds and cross blood–retina barrier in retinopathy. Our work suggests that in vivo phage display combined with microdialysis can be used for the discovery of vascular homing peptides capable of extravasation and tissue penetration.

## Introduction

Targeted drug delivery with affinity ligands such as peptides and antibodies is emerging as an important strategy to improve the therapeutic index of drugs (Ruoslahti, 2022). In vivo phage display has proven useful for identification of the homing peptides that target disease- and organ-specific vascular ZIP codes (Pasqualini & Ruoslahti, 1996; Ruoslahti, 2022). In this agnostic technique, vascular diversity is systemically probed with pools of bacteriophages displaying genetically encoded random peptide libraries (Teesalu et al, 2012).

However, there are additional entry barriers for efficient drug accumulation in the target parenchyma, not just to the vascular wall and its specialized versions (such as blood–brain barrier in the brain), but also to disease-generated barriers such as dense fibrous matrices and high interstitial pressure pumping drugs out of the tumors (Ruoslahti, 2017, 2022, 2024; Ukidve et al, 2021; Vargason et al, 2021). Interestingly, in vivo phage screening has revealed some peptides that not only home to their target vasculature but also extravasate and penetrate the tissue parenchyma (Järvinen & Ruoslahti, 2010; Sugahara et al, 2010; Ruoslahti, 2017, 2022). Such peptides are useful for drug delivery to therapeutically relevant cell populations (Laakkonen et al, 2002; Sugahara et al, 2009; Järvinen & Ruoslahti, 2010; Scodeller et al, 2017) or to extracellular matrix (Lingasamy et al, 2019, 2020) in the tissue parenchyma. Furthermore, their molecular mechanism of tissue penetration has provided insight into molecular recognition events hijacked by pathogenic viruses to penetrate the host tissues and infect the cells (Cantuti-Castelvetri et al, 2020; Daly et al, 2020; Balistreri et al, 2021).

Microdialysis is increasingly applied as an analytical technique for pharmacokinetic and pharmacologic studies in preclinical and clinical settings (O'Connell & Krejci, 2022). In this technique, a fenestrated microprobe covered with a semipermeable membrane is implanted into the site of interest. Subsequently, a buffer is pumped through the probe and compounds under the membrane exclusion limit (separation threshold ranging from low molecular compounds to nanoparticles) in the microenvironment surrounding the probe are sampled. The main advantage of microdialysis is that it allows real-time monitoring of the analytes in the extravascular microenvironment, rather than sampling the blood compartment or the whole tissues (Brunner & Langer, 2006). Standard in vivo phage display protocols do not include elements that favor selection for peptides with tissue-penetrating ability. Currently used modifications of in vivo phage biopanning, such as systemic perfusion with low pH buffer to inactivate phages accessible from systemic circulation (Teesalu et al, 2012), fail to distinguish between phages that accumulate in the endothelial cells and the phages that after initial vascular homing extravasate and enter into the tissue parenchyma.

Here, we report the application of a combination of in vivo peptide phage display and microdialysis (Fig S1A) for discovery of

---

[1]Faculty of Medicine and Health Technology, Tampere University, Tampere, Finland   [2]Department of Orthopedics and Traumatology and Eye Centre, Tampere University Hospital, Tampere, Finland   [3]Laboratory of Precision- and Nanomedicine, Institute of Biomedicine and Translational Medicine, University of Tartu, Tartu, Estonia

Correspondence: tero.jarvinen@tuni.fi
*Toini Pemmari, Stuart Prince, and Niklas Wiss contributed equally to this work

peptides that besides site-specific homing promotes extravasation and tissue penetration. In our proof-of-concept studies on the skin wound model in rats, we performed in vivo biopanning with extravascular phage rescue using a custom-made microdialysis catheter implanted in the wound tissue, and subsequent identification of peptides by high-throughput sequencing (HTS) of the peptide-encoding region of the phage genomic DNA. We show that phage clones displaying homing peptides capable of penetrating wound tissue selectively accumulate the microdialysate and identify new peptides with such properties. This approach should be useful in the discovery of extravasating, tissue-penetrating peptides for other diseased and normal tissues.

## Results

### In vitro capture of bacteriophages by microdialysis probes

T7 bacteriophages, widely used for in vivo biopanning, are biological nanoparticles with a ~60-nm icosahedral head, a noncontractile tail (~20 nm in length and ~10 nm wide), and six thin flexible tail fibers (Granoff & Webster, 1999). We first characterized the ability of T7 phages to cross the microdialysis membranes with a cutoff 1 and 3 MDa (pore size of ~60 and ~200 nm, respectively) (Fig S1B and C). The microdialysis probes were placed in PBS containing $0.8 \times 10^9$ phages/ml and incubated for 1 h, and phage titers in dialysates were determined by the bacterial plaque assay. The dialysate was found to contain $1.3 \times 10^6$ viable phages/ml when the 1-MDa probe was used and $2.9 \times 10^7$ phages/ml with the 3-MDa probe. Thus, ~1 in 27 phages had crossed the membrane into the dialysate from the 3-MDa probe, showing that phage particles do effectively pass through the membrane of the microdialysis catheter.

### In vivo microdialysis and tissue-penetrating homing peptides

We next tested the applicability of the microdialysis-based system for extravascular phage recovery in the rat wound model. In the initial proof-of-concept studies, we used an in vivo "playoff" approach that allows parallel auditioning of the performance of peptide phages in low-complexity mixtures (Pleiko et al, 2021). Wound-bearing rats were dosed with a pool of phages displaying peptides known to home to angiogenic vessels and penetrate tissues, CAR (amino acid sequence CARSKNKDC) and iRGD (CRGDKGPDC). The CAR peptide uses a heparan sulfate proteoglycan (syndecan-4)–dependent pathway for the wound homing and penetration (Järvinen & Ruoslahti, 2007, 2010; Maldonado et al, 2023), whereas iRGD first homes to $\alpha v$ integrins in angiogenic vasculature followed by proteolytic cleavage of the peptide and binging to neuropilin-1, which activates a trans-tissue transport pathway, the CendR pathway (Sugahara et al, 2009, 2010; Teesalu et al, 2009). These receptors are up-regulated during the wound repair process (Matthies et al, 2002; Gopal et al, 2021). Peptides lacking tissue-penetrating properties were included as controls. The CRK (CRKDKC) peptide homes to angiogenic blood vessels, but remains bound to the blood vessels (Järvinen & Ruoslahti, 2007; Agemy et al, 2010). The test pool also included peptide phage clones identified during an in vivo phage display screen for peptides homing to skin wounds but not

selected for further analysis at that time (Järvinen & Ruoslahti, 2007). As negative controls, we included phages from a naïve peptide phage library that display random peptides.

Microdialysis probes with 3-MDa cutoff were implanted under the center of a day 7 skin wound (peak of angiogenesis) in rats, or under normal skin (Fig S2). Care was taken not to rupture blood vessels during the probe insertion, and a proper probe location was confirmed at the end of the experiment. After insertion, the microdialysis device was run for 45 min to allow the local flow inside the catheter to be established (manufacturer's recommendation) (Joukhadar & Müller, 2005; Chaurasia et al, 2007), followed by intravenous injection of the phage cocktail, and sampling of wound and control dialysates for 60 min. Subsequently, the animals were perfused to remove free phages, and wounds and control organs were collected for further analysis. HTS of the peptide-encoding portion of the phage genome, used to assess the presence and representation of each phage, revealed the presence of all of the known wound-homing peptide phages in the wound tissue extract, whereas the microdialysates contained only the phage clones expressing the two peptides with reported cell- and tissue-penetrating properties, CAR and iRGD (Table 1). Notably, the wound-homing peptide CRK that lacks cell and tissue penetration (Järvinen & Ruoslahti, 2007) was absent in the dialysate. No phages (including CAR and iRGD phages) were detected in the dialysates placed in the intact normal skin (Table 1). These results suggest that the microdialysis can be used for selective recovery of phage clones that display peptides capable of extravasation and tissue penetration.

### In vivo phage display screen using microdialysis

We next adopted microdialysis-based biopanning to identify new wound-penetrating peptides. We used a CX7[trinuc]C peptide T7 phage library (diversity $3.9 \times 10^8$) in which random amino acids are encoded by equally represented trinucleotides to prevent codon bias (Pleiko et al, 2021). The library was cloned into the genome of the T7 bacteriophage engineered for longer circulation half-life and reduced liver uptake (Hodyra-Stefaniak et al, 2019). For biopanning, $3.9 \times 10^{10}$ phages (~100 copies of each phage clone) displaying the C7X[trinuc]C library were injected in the tail vein of rats with day 7 skin wounds. Microdialysates were collected from probes placed in wounds or normal skin (Table S1). Most of the phage recovery at the microdialysis probe happened during the first 2 h after library dosing. The wound dialysates yielded almost all phages introduced into the two microdialysis probes, only one phage clone was recovered from normal skin dialysate (Table S1). We observed that most of the peptide phage clones appeared in the microdialysate during the first 2 h after phage dosing (Table S1). To obtain representative tissue biodistribution of phage clones in relation to dialysates, the skin wound–bearing rats with no inserted dialysis probes were dosed with the naïve CX7[trinuc]C peptide library, followed by 30-min circulation, perfusion, collection of organs, and analysis of peptide landscapes. The two dialysates collected during the first 2 h were subjected to HTS (Table S2). Whole tissue samples from the wound with and without the probe (1 + 5 wounds) at 30 min and 5 h, as well as control samples from 14 different tissues (2 × 14) and from normal skin (1 x with and 2 x without probe), and two serum samples at the time of euthanasia (30 min and 5 h after the library injection) were subjected to HTS. HTS revealed no homing-induced enrichment of certain peptide sequences when wound samples were compared to

**Table 1. Wound-homing peptides with tissue-penetrating properties accumulate selectively in microdialysis dialysate from skin wounds.**

| Phage | Wound microdialysate | | Wound without probe | | Liver | | Normal skin microdialysate | Uninjected library sample | |
|---|---|---|---|---|---|---|---|---|---|
| | Count | % | Count | % | Count | % | Count | Count | % |
| CARSKNKDC (CAR) | 17,713 | 94.74 | 1,614 | 7.43 | 29,894 | 11.34 | 0 | 19,598 | 11.6 |
| CRKDKC (CRK) | 0 | 0 | 303 | 1.39 | 17,235 | 6.54 | 0 | 13,666 | 8.1 |
| CRGDKGPDC (iRGD) | 984 | 5.26 | 1,172 | 5.39 | 8,709 | 3.30 | 0 | 7,773 | 4.6 |
| CERESTKIC | 0 | 0 | 3,133 | 14.42 | 93,283 | 35.39 | 0 | 60,067 | 35.5 |
| CKSVKNREC | 0 | 0 | 14,429 | 66.39 | 108,554 | 41.18 | 2,473 | 59,132 | 35.0 |
| CNGSALPVC | 0 | 0 | 0 | 0 | 3,300 | 1.25 | 0 | 4,653 | 2.8 |
| CAELNDGLC | 0 | 0 | 1,082 | 4.98 | 1,851 | 0.70 | 0 | 2,690 | 1.6 |
| CLLGKNNSC | 0 | 0 | 0 | 0 | 785 | 0.30 | 0 | 1,404 | 0.9 |

A custom-made phage library expressing wound (angiogenesis)-homing peptides (CAR, CRK, iRGD), some capable of penetrating cells and tissues (CAR, iRGD) (Järvinen & Ruoslahti, 2007, 2010; Sugahara et al, 2009, 2010; Teesalu et al, 2009; Maldonado et al, 2023) and others (CRK) (Järvinen & Ruoslahti, 2007; Agemy et al, 2010) lacking the penetration capabilities, was created by cloning the peptide sequence to T7 bacteriophage. The library also contained peptide sequences discovered from previous in vivo phage screen toward the skin wound (Järvinen & Ruoslahti, 2007). Excision skin wounds were created in the back skin of 12-wk-old rat, and microdialysis catheters were inserted underneath the wounds and normal skin. The phage library was injected into the tail vein and allowed to circulate for 1 h, whereas microdialysis dialysate was collected. The phages left in the blood were removed by perfusing the rat's circulation, and tissues were harvested for sequencing of phage peptide inserts and microscopic analyses. All phage clones expressing wound-homing peptides present in the library were identified in regular skin wound screening. In contrast, the microdialysis procedure revealed only the two clones expressing skin-homing peptides with cell- and tissue-penetrating properties (CAR and iRGD).

healthy control organs, which is in line with our previous experience after one round of in vivo phage screening (Table S2). However, the wound microdialysates contained 13 enriched peptide sequences (Table 2). When these sequences enriched at both dialysates were compared to the sequences from controls, it was found that one sequence (CKKNEINNC) was found in one skin wound sample, but all 13 were not detected in control organs and demonstrated enrichment over their presentation in the library (Tables 2 and S2). A BLAST analysis on candidate peptide sequences was used to scan for sequence homologies with proteins related to angiogenesis, endocytosis, and wound healing (Table S3).

Based on the presence of multiple clones at both microdialysis analysis time points, and on sequence similarities to proteins involved in relevant pathways, we selected 3 of the enriched peptides (CDD [CDDYQQISC], CPK [CPKKHHLDC], and CYH [CYHDTYPNC]) for further analysis.

## Cell-penetrating and wound-homing properties in microdialysis-captured peptides

Short homing peptides are generally thought to be not species-specific as they target functionally important binding pockets highly conserved between species on target molecules. We next characterized the cellular uptake and in vivo biodistribution of the synthetic CDD, CPK, and CYH peptides labeled with 6-carboxyfluorescein (FAM) on cultured Chinese hamster ovary (CHO-K) and human endothelial cells (HUVECs) in vitro and in the mouse wound model in vivo. All three peptides bound to and were taken up by CHO-K and HUVECs, as demonstrated by confocal imaging (Figs 1 and S3).

We next studied homing of systemically administered CDD, CPK, and CYH peptides in mouse skin excision wounds at the peak of neo-vascularization, 7 d after wounding. The peptides were injected into the tail vein and allowed to circulate for 2.5 h. The systemically administered peptides were detected in the granulation tissue of the wounds outside of the blood vessels (Figs 2 and S4). Quantification showed that all three homed to skin wounds in significant amounts, the CYH peptide accumulated in wounds ninefold ($P = 0.01$), the CDD 12-fold ($P = 0.02$), and CYH fivefold ($P = 0.03$) more than a control (Fig 3A). In contrast, the peptides were undetectable in normal skin (Fig S5A), heart, liver, lung, and spleen (Fig S5B), except for strong signal related to peptide secretion from the kidney glomeruli (Fig S5B).

Next, we wanted to study the utility of the newly identified peptides as vehicles for targeting diabetic skin wounds. Compromised angiogenesis in diabetic wounds leads to deficient granulation tissue formation and poses an important hindrance to systemic drug delivery. BALB/cJRj and diabetic male BKS(D)-Lepr[db]/JOrlRj mice were wounded and systemically dosed with fluorophore-labeled peptides 7 d after the skin wounding. For all tested peptides, we observed peptide accumulation in the granulation tissue where angiogenesis takes place (Fig S6).

## Homing to retinopathy and penetration of the blood–retina barrier

We wanted to explore whether the new wound-penetrating peptides externalize and penetrate tissues that have an additional barrier around the blood vessels. The retinal blood vessels have an additional element, blood–retina barrier (BRB) that hinders the access of systemically administered drugs to retina (Lahdenranta et al, 2007; Sweeney et al, 2019; Vähätupa, et al, 2020b). Thus, we explored the homing of the new peptides to the normal retina and to the retinal angiogenesis in the oxygen-induced retinopathy (OIR) model (Vähätupa, et al, 2020b). None of the peptides homed to the normal retina (Figs 3B and 4A), but a strong homing signal was detected in OIR retinas around retinal blood vessels and the preretinal blood vessels (tufts) with the new homing peptides, CDD, CPK, and CYH, but not with a control peptide (Fig 4B and C). There was

**Table 2.** Screening of a naïve peptide library by microdialysis-assisted in vivo phage display yields wound-enriched peptide sequences.

| Peptide | First hour (counts) | Second hour (counts) | Total counts |
|---------|---------------------|----------------------|--------------|
| **CPKKHHLDC** | 19 | 23 | 42 |
| **CYHDTYPNC** | 14 | 14 | 28 |
| CNYLVEKNC | 14 | 9 | 23 |
| **CDDYQQISC** | 10 | 11 | 21 |
| CLSQTYRIC | 19 | 2 | 21 |
| CKMLYEYHC | 1 | 17 | 18 |
| CLLIYNWSC | 14 | 3 | 17 |
| CKKNEINNC | 6 | 10 | 16 |
| CLTVLSEQC | 8 | 7 | 15 |
| CYWEDKNLC | 6 | 9 | 15 |
| CFCQLMYQC | 8 | 5 | 13 |
| CRMKFYSEYC | 2 | 9 | 11 |

A custom-made, long-circulating C7X$^{trinuc}$C T7 bacteriophage library with a diversity of $3.9 \times 10^8$ peptides was injected into a rat tail vein. The microdialysate from a 7-d-old wound was collected during the first 2 h after library injection, peptide-encoding inserts in phage DNA were analyzed with Ion Torrent HTS, and the insert sequences were compared with inserts from control organs harvested from the same animal. The frequency of clones detected is shown. The peptides chosen for further analysis are bolded.

also a strong accumulation of three peptides in the OIR vitreous (Fig 4B and C), whereas almost no control peptide was detected in the vitreous (Fig 4B and C). Next, we quantified the area of positive staining in the retina. There was ninefold more of CYH ($P < 0.01$) and 120-fold and 88-fold more of CDD and CPK than of a control peptide in OIR retinas ($P < 0.0001$ for the CDD and CPK peptides) (Fig 3C). The peptide accumulation in the vitreous of the OIR retina was $16.9 \times$ ($P = 0.0005$), $67.2 \times$ ($P < 0.0001$), and $80.3 \times$ ($P = 0.0002$) higher than that of the control peptide for CYH, CDD, and CPK peptides, respectively (Fig 3D).

## Discussion

The vascular system displays significant heterogeneity in its blood vessels (Ruoslahti, 2017, 2022, 2024; Vanlandewijck et al, 2018; Kalucka et al, 2020). This heterogeneity can be exploited to obtain organ- or disease-selective targeting of pharmaceutics by employing homing ligands (Ruoslahti, 2022, 2024). The goal of the current study was to show that adding a microdialysis procedure to the traditional in vivo biopanning provides an efficient screening method for identifying homing peptides that penetrate the target tissue. We demonstrate that T7 bacteriophages can be captured by a custom-made microdialysis probe in vitro that the method distinguishes between known cell- and tissue-penetrating homing peptides and homing peptides lacking penetrating properties. Finally, the method yields new tissue-penetrating homing peptides.

Although the heterogeneity in the vasculature offers unique opportunities for organ- or disease-selective delivery of pharmaceutical agents, there are additional entry barriers for efficient drug accumulation, such as the normal vessel wall, the blood–brain barrier in the brain and the BRB in the retina (Vähätupa, et al, 2020b; Banks et al, 2024; Furtado et al, 2024; Xie et al, 2024), dense fibrous matrices and high interstitial pressure in tumors, and compromised vascular supply in diseases like diabetes (Ukidve et al, 2021; Vargason et al, 2021). The enhanced permeability retention (EPR) effect (Nel et al, 2017; Ukidve et al, 2021) is often thought to overcome these obstacles in tumors because of leakiness of angiogenic tumor vessels (Nel et al, 2017). Our results show that combining in vivo phage display with microdialysis separates phage clones with homing and cell/tissue-penetrating properties from the usual homing clones in the angiogenic vasculature setting. These results suggest that the vessel leakiness does not equate tissue penetration. Effective tissue penetration is likely to require that the peptide is able to use existing trans-tissue transport pathways for spreading in tissue, as is known to be the case with the already known tissue-penetrating peptides (Järvinen & Ruoslahti, 2007; Ruoslahti, 2017, 2022; Pemmari et al, 2020a; Maldonado et al, 2023).

In vivo phage display has been a powerful tool to exploit the heterogeneity in the vasculature and to identify organ- and disease-selective homing peptides. Despite its considerable success, the technology is not without limitations; namely, in vivo phage screen is laborious and subject to several biases and artifacts (Järvinen, 2012; Pemmari et al, 2020b). To overcome these limitations, we developed an approach for identification of target-selective homing peptides based solely on in vivo display HTS data. However, even the introduction of HTS for the analysis of in vivo display data could not shorten the need for multiple rounds of screening to identify homing selectivity (Pleiko et al, 2021). The multiple screening rounds, in turn, are not only laborious, but more importantly expose the random library to bias related to peptide-dependent variation in the phage amplification rates between the screening rounds and propensity of phage to induce mutations that cause library peptides expressed as the extension of the coat protein not to be expressed at all (Pleiko et al, 2021). To overcome these biases in the screen, we demonstrate that the additional selection step introduced by the microdialysis probe allows the identification of novel homing peptide sequences capable of target tissue homing and tissue penetration after one screening round. This not only cuts the amount of work and expenses, but more importantly identifies a functional property, cell/tissue penetration, while also reducing the potential intrinsic bias introduced to the screen by uneven amplification of phage clones between the screening rounds or complete disappearance of the expressed peptides that lead to skewed representation of peptide sequences in the screened library.

All the three peptides chosen for further analysis share homology with proteins involved in angiogenesis, skin wound healing, and cell penetration. The CPK (CPKKHHLDC) peptide sequence has almost complete homology with sequence in the AP180 N-terminal homology (ANTH) domain in the clathrin coat assembly protein AP180 and its nonneuronal homolog phosphatidylinositol-binding clathrin assembly protein (PICALM) for cell internalization (McMahon, 1999). Clathrin-mediated endocytosis is the major pathway for receptor-mediated endocytosis, estimated to contribute to over 95% of endocytic flux (Bitsikas et al, 2014). PICALM has recently been shown to immunoprecipitate together with the major angiogenic growth factor receptor, VEGFR2, in endothelial cells (Ho et al, 2024). The CYH (CYHDTYPNC) peptide has homology with the protein sequence in the extracellular domain of the cell membrane–located NRP-like protein discoidin. Discoidin interacts with VEGFR2 and promotes VEGF signaling

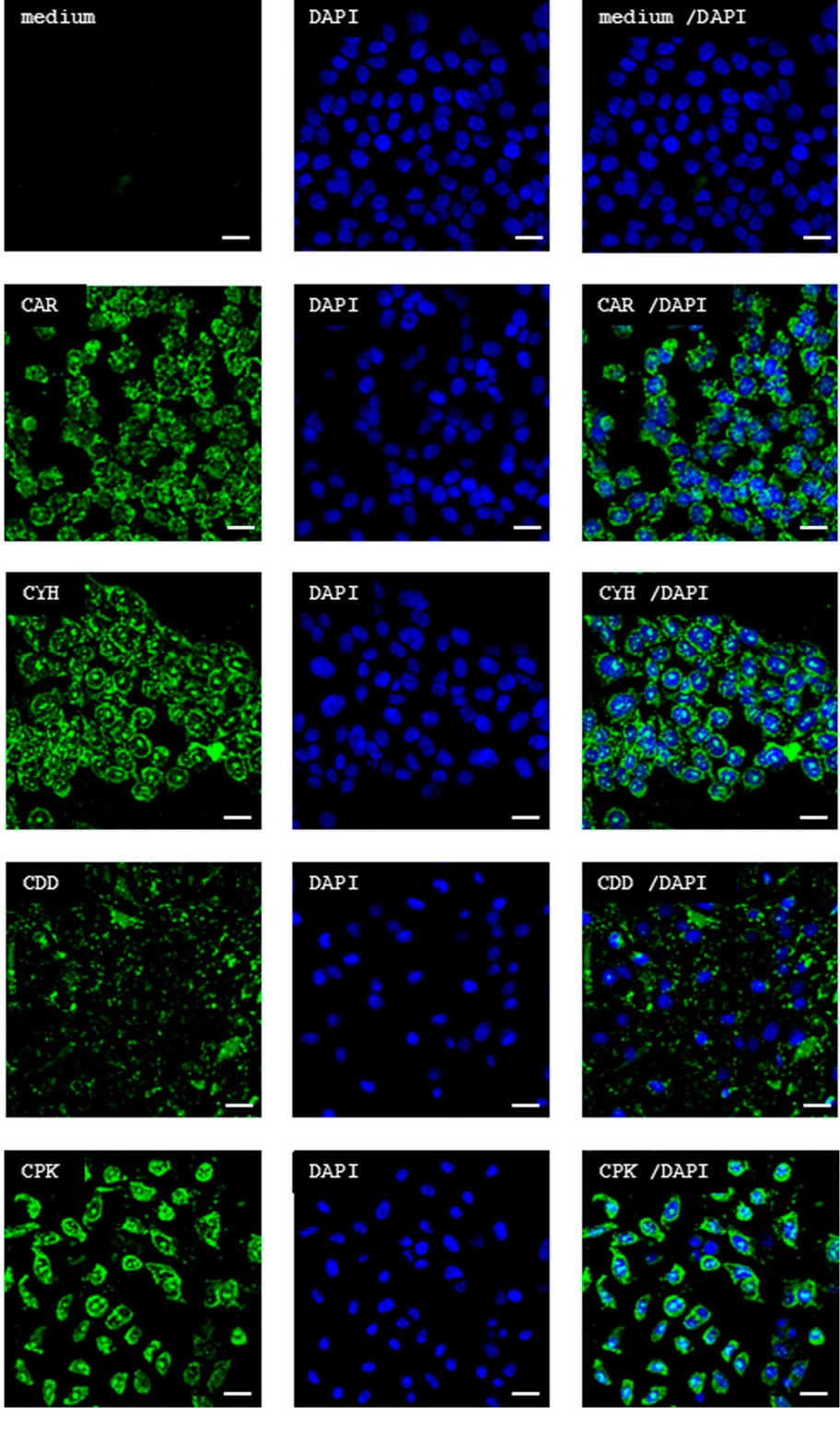

**Figure 1. Microdialysis-captured peptides penetrate the cell membrane.**
The three novel peptides (CDD, CPK, and CYH) identified from in vivo phage display screen performed in combination with microdialysis accumulate in the CHO-K cells as does CAR peptide, previously shown to internalize these cells. CHO-K cells were grown on chamber slides. The FAM-labeled CAR, CDD, CPK, or CYH peptide (green, 20 µM) or medium (vehicle) was added to the cells, and the cells were incubated for 3 h. After that, the chamber slide was rinsed, fixed, and stained with DAPI (blue). The cells were imaged with a Zeiss LSM780 inverted confocal microscope. The scale bar is 20 µm. Images are representative of three individual experiments.

(Kobuke et al, 2001; Sadeghi et al, 2007; Guo et al, 2009; Nie et al, 2013). The CDD (CDDYQQISC) peptide bears resemblance to the sequence in the leucine-rich repeat region D1 in the slit homolog 2 protein, which is endothelium-derived and plays a role in angiogenesis and cell migration (Wu et al, 2001). These homologies strongly suggest that our new tissue-penetrating peptides use physiological transport mechanisms to penetrate and spread in tissues. Further analysis of the mechanisms involved should prove rewarding.

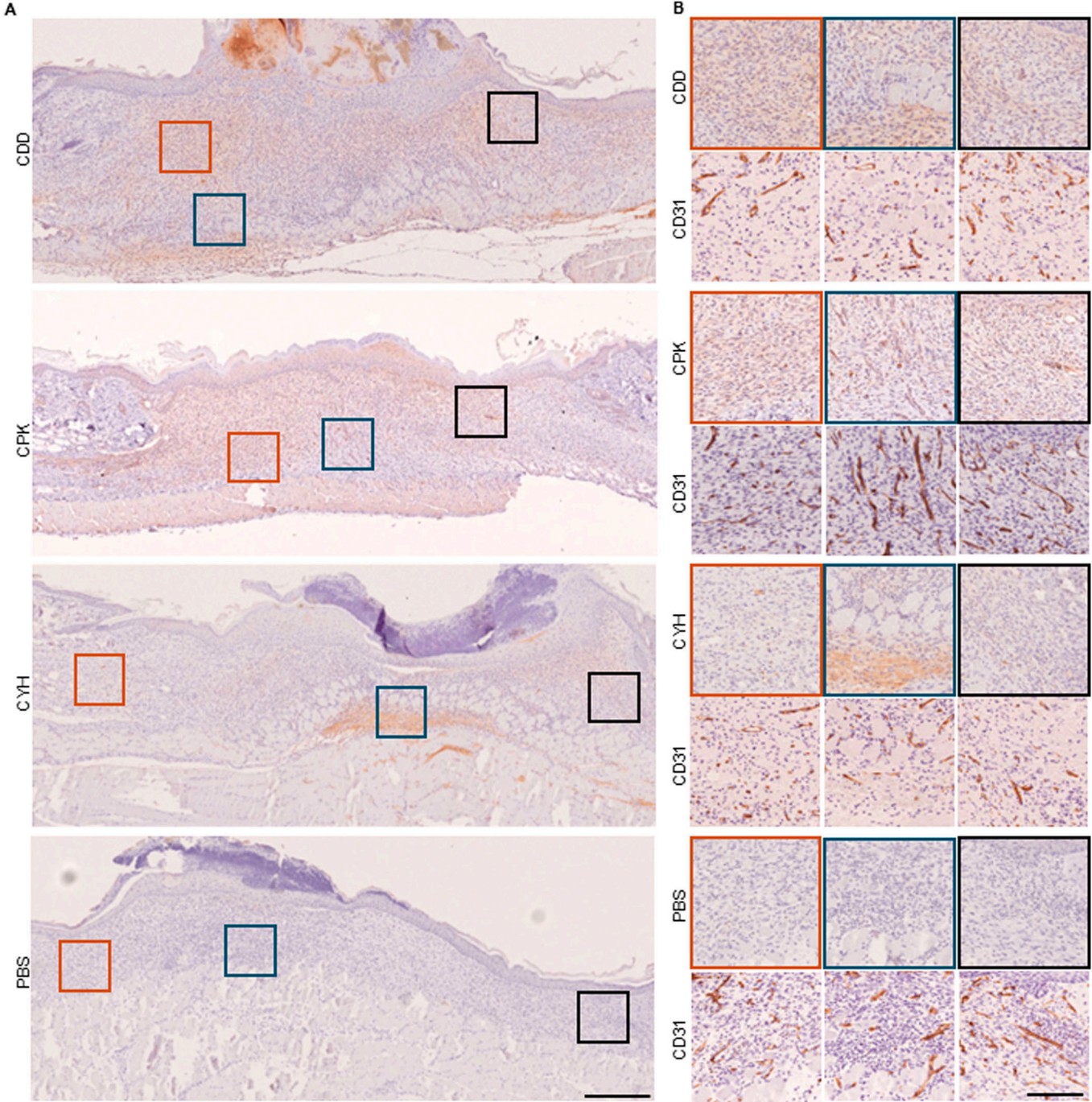

**Figure 2. Microdialysis-captured peptides home to skin wounds and localize in granulation tissue.**
The homing peptides enriched in the microdialysis screens (CDD, CPK, and CYH) accumulated in wound regions containing blood vessels and in surrounding extravascular granulation tissue. Six-mm-diameter full-thickness wounds were cut in the back skin of 8-wk-old male mice. After 7 d, the peptides or PBS was injected systemically (tail vein). The peptides were allowed to circulate for 150 min before the vasculature was thoroughly perfused. The wound tissue was fixed, and microscopic sections were stained with anti-FAM and hematoxylin. **(A, B)** Magnifications of areas marked in panel (A) are represented in panel (B) with adjacent sections stained with anti-CD31 to visualize the blood vessels. The scale bar is 500 μm for panel (A) and 125 μm for panel (B). Images are representative of two separate experiments, four mice in each peptide group.

The previously known tissue-penetrating peptides have come from large numbers of in vivo phage screens where only a very small minority of the peptides displayed such properties. In contrast, we identified several such peptides in a limited number of screens employing microdialysis. Applying this method to tissues other than the two analyzed in this study may greatly expand the repertoire of homing peptides that can effectively deliver drug or diagnostic payloads to the extravascular compartment of disease tissues.

# Materials and Methods

**Structured methods**

**Reagents and tools table.**

| Reagent/resource | Reference or source | Identifier or catalog number |
|---|---|---|
| Experimental models | | |
| BALB (*M. musculus*) | Janvier Labs | BALB/cByJRj |
| C57BL/6Rj (*M. musculus*) | Janvier Labs | C57BL/6JRj |
| BKS(D)-Lepr<sup>db</sup> (*M. musculus*) | Janvier Labs | BKS(D)-Lepr<sup>db</sup>/JOrlRj |
| HUVEC (*H. sapiens*) | Promo Cell | Cat # C-12200 |
| Chinese ovary (CHO-K1, *hamster*) | ATCC | Cat # CCL-61 |
| Novagen T7 select system | Merck Millipore | Cats # 70015 and 70550 |
| Wistar (*R. norvegicus*) | Janvier Labs | RjHan:WI |
| Recombinant DNA | | |
| Phage library | This study | NA |
| Antibodies | | |
| Anti-rabbit Alexa Fluor 594 | Jackson | Cat # AB_2340621 |
| DAB | Agilent | Cat # K3468 |
| Rabbit anti-FITC | Thermo Fisher Scientific | Cat # 71-1900 |
| Rabbit anti-VE-cadherin | Abcam | Cat # ab205336 |
| Rat anti-CD31 | BD Pharmingen | Cat # 550274 |
| Oligonucleotides and other sequence-based reagents | | |
| FAM-labeled peptides | This study | Table 2 |
| Chemicals, enzymes, and other reagents | | |
| AMPure XP Bead–Based Next-Generation Sequencing Cleanup system | Beckmann Coulter | Cat # A63881 |
| Attane vet | Piramal Critical Care B.V. | NA |
| Biocare Medical Rabbit on Rodent | Biocare Medical | Cat # RMR622H |
| BSA | Sigma-Aldrich | Cat # A4612 |
| Norocarp | Norbrook Laboratories | NA |
| CMA perfusion fluid | CMA Microdialysis AB | Cat # CMAP000151 |
| DetachKit | PromoCell | C-41200 |
| Domitor vet | Orion Corporation | NA |
| DMEM | Gibco | Cat # D6546 |
| Endothelial Cell Growth Medium 2 | PromoCell | Cat # C-22011 |
| Ethanol | Anora | Cat # 1025902 |
| Hematoxylin | Merck | Cat # 1.09253 |
| High-sensitivity DNA Kit | Agilent | Cat # 5067-4626 |
| Histofine Simple Stain Mouse MAX PO | Nichirei | Cat # 414311F |
| Ion PGM HiQ View OT2 Kit | Life Technologies | Cat # A29900 |
| Ion PGM HiQ View Sequencing Kit | Life Technologies | Cat # A30044 |
| Ketalar | Pfizer | NA |
| Normal goat serum | Thermo Fisher Scientific | Cat # 31872 |

**Continued**

| Reagent/resource | Reference or source | Identifier or catalog number |
|---|---|---|
| NP-40 | Thermo Fisher Scientific | Cat # 28324 |
| Paraffin | Sigma-Aldrich | Cat # 17310 |
| PFA | Sigma-Aldrich | Cat # 818715 |
| PBS | Thermo Fisher Scientific | Cat # 10010023 |
| Phusion Green Hot Start II High-Fidelity DNA Polymerase | Thermo Fisher Scientific | Cat # F537L |
| Sevoflurane Baxter | Baxter SA | NA |
| Tween-20 | Thermo Fisher Scientific | Cat # 85113 |
| VECTASHIELD Antifade Mounting Medium with DAPI | Vector Laboratories | Cat # H-1200-10 |
| Vetergesic | Ceva Santé Animale | NA |
| XM-Factor | Biocare Medical | Cat # XMF963C |
| Software | | |
| Basic Local Alignment Tool | Altschul et al (1990) | https://blast.ncbi.nlm.nih.gov |
| edgeR | Robinson et al (2010) | http://bioconductor.org/packages/release/bioc/html/edgeR.html |
| UniProt | UniProt Consortium (2019) | https://www.uniprot.org |
| QuPath | Bankhead et al (2017) | https://qupath.github.io |
| Other | | |
| Agilent Bioanalyzer 2100 | Agilent | Cat # G2939BA |
| Biopsy punch | Kai medical | Cat # BP-60F |
| CMA 1-MDa probe | CMA Microdialysis AB | Custom-made |
| CMA 3-MDa probe | CMA Microdialysis AB | Custom-made |
| CMA 402 Syringe Pump | CMA Microdialysis AB | Cat # 8003100 |
| CMA connector tubing | CMA Microdialysis AB | Cat # 3409500 |
| CMA FEP tubing | CMA Microdialysis AB | Cat # 3409501 |
| CMA guide cannula | CMA Microdialysis AB | Cat # 8309024 |
| CMA Probe Shaft Clip | CMA Microdialysis AB | Cat # CMA8309003 |
| Contour Next One | Ascensia Diabetes Care | Cat # 85213326 |
| Harvard P70 peristaltic pump | Harvard Apparatus | Cat # 70-7000 |
| Ion 316v2 chip | Life Technologies | Cat # 448149 |
| Ion Torrent Personal Genome Machine | Thermo Fisher Scientific | Cat # 4462921 |
| Menzel 1-mm Microscope Coverslip | Thermo Fisher Scientific | Cat # 10083957 |
| NanoZoomer S60 | Hamamatsu Photonics | Cat # C13210-01 |
| Nunc Lab-Tek II Chamber Slide system | VWR | Cat # 734-2050 |
| Syringe (1 ml) | Hamilton Company | Cat # P209681 |
| Zeiss LSM780 Laser Scanning Confocal Microscope | Carl Zeiss AG | NA |
| ProOx P110 oxygen controller | BioSpherix Ltd | NA |
| NanoZoomer S60 | Hamamatsu Photonics | NA |

## Phage and peptide synthesis

Three different peptide libraries expressed in bacteriophage were used. Novagen T7 Select System (Merck Millipore) was used in all of them according to the instructions provided by the manufacturer.

For the in vitro validation, a C7XC T7 phage library with 415-1 backbone (415 peptide copies per phage) grown in BL21 *Escherichia coli* was used. The titer of the library was $0.8 \times 10^9$ per ml.

The custom-made phage library for the in vivo validation was cloned on T7 415-1 backbone, grown in BL21 *E. coli* strain, and filter-

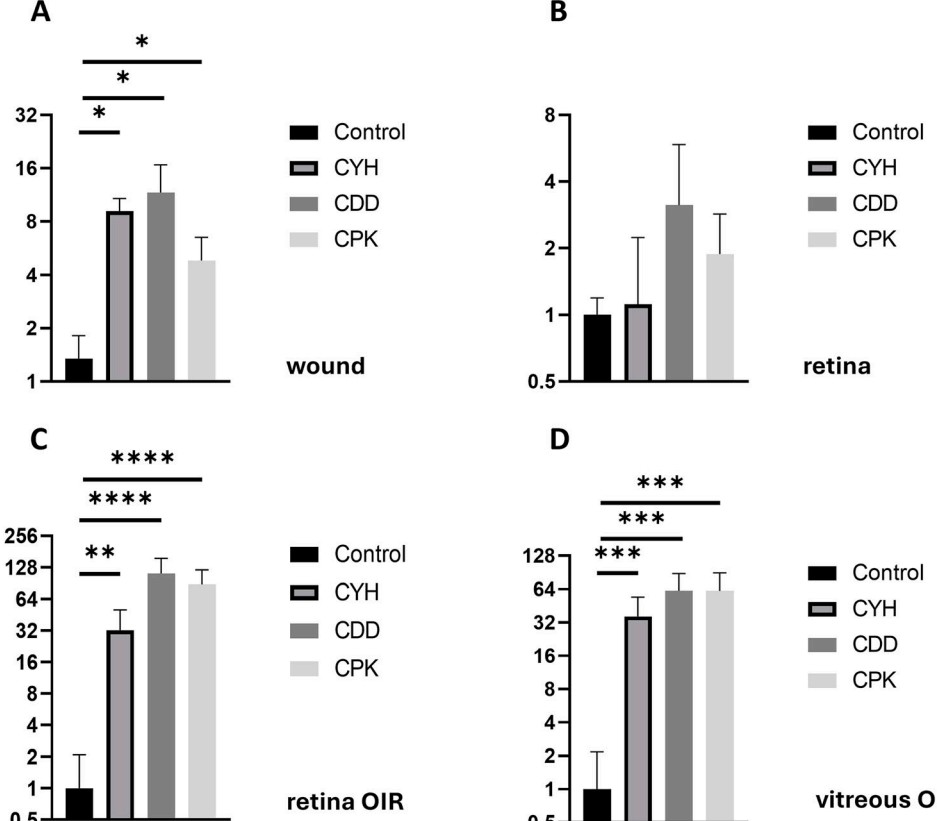

**Figure 3. Microdialysis-captured peptides home to skin wounds and penetrate blood–retina barrier in retinopathy.**
Systemically administered homing peptides CDD, CPK, and CYH accumulate in skin wounds and penetrate blood–retina barrier in retinopathy. Full-thickness wounds, 6 mm in diameter, in the back skin of 8-wk-old male mice were used. After 7 d, each peptide or PBS was injected into the tail vein. For retinopathy experiments, oxygen-induced retinopathy (OIR) was induced by placing mouse pups at day P7 in a hyperoxia chamber for 5 d. On P17, when the angiogenesis peaks in this OIR model, the peptides were injected into the tail vein. The peptides circulated for 150 min (wounds) and 120 min (OIR) before the vasculature was thoroughly perfused. The skin wounds and eyeballs were fixed, and histological samples stained with anti-FAM and hematoxylin. Peptide homing was quantified with the automated image analysis system. **(A)** Microdialysis-identified peptides home to skin wounds. **(B, C)** Peptides do not home to the normal retina (B), but significant homing is detected in the retina in the OIR model (C). **(D)** All three microdialysis-identified peptides also accumulate substantially in the vitreous in the OIR model. *$P < 0.05$, **$P < 0.01$, ***$< 0.001$. Each datapoint represents an individual wound, retina, or vitreous. N = 3 for skin wound, and N = 6 and N = 8 for normal and OIR retinas.

sterilized. The peptide sequences used in the in vivo validation are listed in Table 1. The phages were either cloned as described previously or picked randomly from a phage library (Järvinen & Ruoslahti, 2007; Teesalu et al, 2009; Pemmari et al, 2020a). Each phage clone was sequenced. The custom-made library was made by mixing an equal amount of each phage (titer $5 \times 10^9$).

The phage library used in the in vivo screen with the microdialysis probe was in the 10-3b T7 backbone (5–15 peptide copies per phage), grown in BLT5403 *E. coli* strain, CsCl-purified, dialyzed in PBS, and filter-sterilized. The library was constructed via trinucleotide synthesis to prevent bias in the amount of different amino acids (Kayushin et al, 1996). The diversity of the library was $1 \times 10^9$ and the titer $5 \times 10^{10}$ per ml.

Peptides were synthesized with an automated peptide synthesizer using standard solid-phase fluorenylmethoxycarbonyl chemistry. During synthesis, the peptides were labeled with fluorescein amidite (FAM) using an aminohexanoic acid spacer as previously described (Laakkonen et al, 2004).

### Microdialysis equipment

For the in vitro validation, CMA 402 Syringe Pump (CMA Microdialysis AB) with two Hamilton 1-ml syringes (Hamilton Company) was used to push and pull the perfusion fluid. The push syringe was connected to the inlet spike of the CMA 12 custom-made 3-MDa probe

or CMA 12 1-MDa probe (both from CMA Microdialysis AB) and the outlet spike to the pull syringe via CMA FEP tubing (CMA Microdialysis AB).

For the in vivo microdialysis, CMA 402 Syringe Pump with Hamilton syringes was used to push and the Harvard P70 peristaltic pump (Harvard Apparatus) to pull the CMA perfusion fluid (CMA Microdialysis AB) and dialysate. Each of the CMA 402 two outputs was connected to the inlet spike of the CMA 12 custom-made 3-MDa probe (CMA Microdialysis AB) via CMA FEP tubing (CMA Microdialysis AB), and the outlet spike of the CMA 12 probe was further connected to Harvard P70 peristaltic pump and further to CMA Probe Shaft Clip (CMA Microdialysis AB) via CMA connector tubing (CMA Microdialysis AB). From the CMA Probe Shaft Clip, the dialysate was pumped into CMA sample tubes (CMA Microdialysis AB). Harvard 0.19-mm-diameter tubing (Harvard Apparatus) and Harvard 3 collar 1.52-mm tubing (Harvard Apparatus) were used with the Harvard P70 peristaltic pump. In addition, CMA tubing adaptors (CMA Microdialysis AB) were used for connecting tubes to syringes and probes. The system was flushed with 70% ethanol before any use.

### In vitro microdialysis

Before starting the experiment, the tubing system was cleaned by flushing two circuit volumes of autoclaved 0.1 M glycine (pH 3), 70% EtOH, ultrapure water, and finally autoclaved Ringer's solution. The

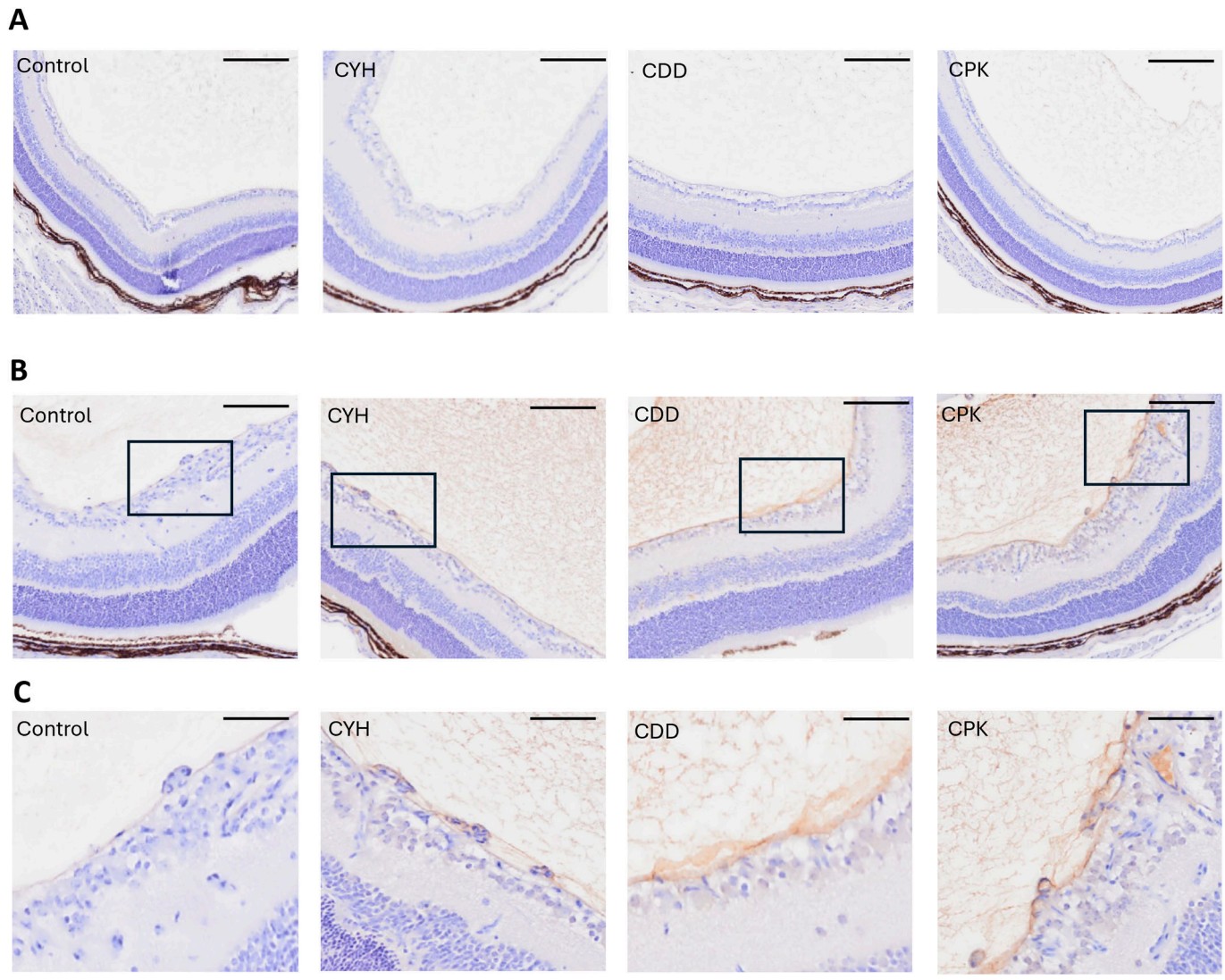

**Figure 4. Microdialysis-captured peptides home and penetrate to retina in retinopathy.**
The CDD, CPK, and CYH peptides accumulate in the retina and vitreous in oxygen-induced retinopathy (OIR). The OIR model was carried out as described previously (Vähätupa et al, 2020b). At P17, the peptides were injected into the tail vein and circulated for 120 min before the vasculature was thoroughly perfused. The eyeballs were fixed and histological sections stained with anti-FAM antibody and hematoxylin. **(A)** None of the tested peptides homed to the normal retina after the systemic administration at P17. **(B)** Three microdialysis-identified homing peptides homed and penetrated into the retina in OIR at the peak of angiogenesis at P17. The peptides also accumulate in the vitreous in the OIR model. **(A, B, C)** High-magnification images of peptide accumulation in OIR retina. The scale bar is 100 μm for (A, B) and 20 μm for (C). Images are representative of N = 6 and N = 8 for normal and OIR retinas.

pull syringe was emptied and run with 100 μl sterile Ringer's solution. The connected CMA 12 probe tip was suspended in the C7XC library, and the dialysis was run using Ringer's solution as perfusion fluid.

## Animals and wound models

Male Wistar rats were used for the in vivo validation experiment and the library screen. The rats were 27 wk old and weighed 490 g. Two full-thickness circular wounds were created on a shaved and disinfected skin with a 12-mm-diameter biopsy punch and scissors under sevoflurane anesthesia (Sevoflurane Baxter, Baxter SA) and carprofen analgesia (50 mg/kg; Norocarp, Norbrook Laboratories).

Seven days after the wounding, the microdialysis experiment was performed. The animal use was accepted by the National Animal Ethics Committee of Finland on 13 November 2013 (ESAVI/6330/04.10.07/2013) and 12 September 2017 (ESAVI/6422/04.10.07/2017). The experiments were conducted according to the ARRIVE guidelines.

Male BALB/cJRj and diabetic male BKS(D)-Lepr[db]/JOrlRj mice were used for the peptide homing experiment. The BALB/cJRj mice were 8 wk old and weighed 24–28 g. The BKS(D)-Lepr[db]/JOrlRj mice were 14 wk old and weighed 45–58 g, and their blood sugar levels were over 33 mmol/liter. Four full-thickness circular wounds were created on a shaved and disinfected skin with a 6-mm-diameter biopsy punch and scissors under isoflurane

anesthesia (Attane vet 1,000 mg/g; Piramal Critical Care B.V.) and buprenorphine analgesia (0.15 mg/kg; Vetergesic vet 0.3 mg/ml; Ceva Santé Animal). Seven days after the wounding, the homing experiment was performed. The animal use was accepted by the National Animal Ethics Committee of Finland on 12 September 2017 (ESAVI/6422/04.10.07/2017) and on 4 January 2022 (ESAVI/39175/2021).

### Oxygen-induced retinopathy (OIR)

WT C57BL/6Rj mice were used for the OIR experiments (Vähätupa, et al, 2020a). The OIR model was performed as described in detail previously (Vähätupa, et al, 2020a; De Rossi et al, 2021). Briefly, to induce retinopathy in the retina, the pups and their nursing mothers were exposed to 75% oxygen in a custom-made chamber (ProOx P110 oxygen controller; BioSpherix Ltd.) at postnatal day 7 (P7) for 5 d until P12 when they were returned to normal room air. The homing experiments were performed at P17 (late hypoxic phase and the peak of neovascularization) (Vähätupa et al, 2021). Control animals were housed under normal room air conditions, and retinas were harvested on the corresponding day (P17). All retina experiments were conducted under the ARVO Statement for the Use of Animals in Ophthalmic and Vision Research guidelines. The animal use for the OIR model was accepted by the National Animal Ethics Committee of Finland on 22 October 2020 (ESAVI/22779/2020).

### In vivo microdialysis

The tubing system was cleaned as described above. A 12-wk-old male rat with two 7-d-old wounds was anesthetized with inhaled sevoflurane. A small skin incision was made ~16 mm from the edge of the wound, and a 16-mm CMA guide cannula (CMA Microdialysis AB) was inserted and superglued in place in the fashion that the cannula ended at the edge of the wound. The metal pin was replaced by a CMA 12 custom-made 3-MDa probe. The microdialysis membrane extends 12 mm out of the cannula into the wound with a 12 mm diameter. Similarly, another CMA guide cannula and a CMA 12 custom-made 3-MDa probe were placed in normal skin far from wounds. The pump system was run at 0.5 $\mu$l/min for 45 min. (If bleeding took place, i.e., dialysate turned red, the experiment was stopped.) Then, 200 $\mu$l of the custom-made library or 2 ml of the screening library was injected via the tail vein. The first 30 min of collected dialysate was discarded as dead volume. The dialysate was collected for 1 h (in vivo validation) or at timed intervals described in Table S1 (library screen). When the dialysis was ready, the rat was anesthetized with subcutaneous medetomidine–ketamine injection (0.4 mg/kg, Domitor vet 1 mg/ml; Orion Corporation, and 60 mg/kg, Ketalar 50 mg/ml; Pfizer Oy) and thoroughly perfused with sterile 1% BSA in DMEM as described previously (Järvinen, 2012). The dialysate from skin wound and normal skin, a sample of the injected library as well as the wound with the probe (proper probe placement and no visible hematoma around the probe were checked), the wound without a probe, and tissue pieces from liver and normal skin without a probe were collected as samples. In the library screen, the following samples were collected in addition to the aforementioned samples: eyes, Achilles and patellar tendons,

gastrocnemius and soleus muscles, heart, kidney, spleen, and tissue pieces from liver, bladder, colon, lung, and small intestine.

### Determination of phage-displayed peptide sequences

The tissue samples harvested from animals were snap-frozen and later broken down by a hand-held homogenizer and suspended in LB/1% NP-40. The peptide-encoding region of the bacteriophage genome was amplified by PCR using Phusion Green Hot Start II High-Fidelity DNA Polymerase (F537L; Thermo Fisher Scientific) (reaction volume: 25 $\mu$l). Cycling conditions were as follows: denaturation at 98°C for 30 s, followed by 25 amplification cycles (10 s at 98°C, 21 s at 72°C), and final elongation (72°C for 5 min). PCR products were purified using the AMPure XP Bead Based Next-Generation Sequencing Cleanup system (A63881; Beckmann Coulter) using 1,8 $\mu$l of beads per 1 $\mu$l of PCR products. Purified PCR products were quantified using Agilent Bioanalyzer 2100 Instrument using High-sensitivity DNA Kit (5067-4626; Agilent). Ion Torrent Emulsion PCR and enrichment steps were performed using Ion PGM HiQ View OT2 Kit (A29900; Life Technologies). HTS was performed using Ion Torrent Personal Genome Machine (Ion PGM) using Ion PGM HiQ View Sequencing Kit (A30044; Life Technologies) and Ion 316v2 chips (448149; Life Technologies). The FASTQ sequence files were converted to text files and translated using in-house–developed Python scripts.

### Peptide data mining

edgeR (Robinson et al, 2010) was used for normalization of peptide count in the samples and for statistical analyses. Peptides with less than five reads present in less than two replicate samples were not used for further analysis. Peptides present in blood, dialysate, skin, and wound samples versus input library were compared for peptide representation analysis and with the peptides present in the control organs (listed under the In vivo microdialysis heading). A false discovery rate (FDR) cutoff of 0.05 and a logFC cutoff of 2 were used as thresholds for statistical significance.

### BLAST analysis

Basic Local Alignment Tool (BLAST) (Altschul et al, 1990) was used to search for counterparts of the peptide sequences enriched in the library screen from human proteome. The protein–protein part of the software was used with the following settings: Database: Nonredundant protein sequences (nr), Organism: human (tax id: 9606), Algorithm: blastp, Max target sequences: 100, Expected threshold: 200,000, Word size: 2, Matrix: PAM30, and no Compositional adjustment and the Short queries option was chosen. All 12 peptide sequences in Table 2 were blasted without the initial and final cysteines. The results were arranged according to Query Cover percentages, and the found proteins were searched from the rat (*Rattus norvegicus*) and mouse (*Mus musculus*) proteome using the UniProt database (UniProt Consortium, 2019). Only the proteins found from all three proteomes were included in the results. The subcellular locations of the proteins were acquired from the UniProt database.

## Peptide internalization in vitro

CHO-K cells were obtained from the American Type Culture Collection (ATCC). Cells were maintained in α-MEM and Earle's salt supplemented with 10% FBS, 100 μg/ml streptomycin sulfate, 100 U/ml of penicillin G, and 292 μg/ml L-glutamine (Thermo Fisher Scientific). HUVECs (PromoCell) were grown as a monolayer in Endothelial Cell Growth Medium 2 (Promo cell). When the cells were about to reach confluence, they were subcultured to eight-well Nunc Lab-Tek II Chamber Slide (Thermo Fisher Scientific) in the growth medium. After 3–4 d of incubation, peptides CAR, CDD, CPK, and CYH were added. The peptides were diluted in PBS to 4 mM concentration, and 1 μl of this solution was used to add the peptides to the growth medium to a final incubation concentration of 10 μM in 400 μl growth medium. Plain growth medium was used as a negative control. The chamber slides were incubated for 3 h at 37°C, 5% $CO_2$.

After the incubation, the chamber slide was rinsed three times with cell medium and the cells were fixed with 2% PFA for 10 min. The HUVECs were then rinsed two times with 0.05% Tween in PBS, and 0.2% Tween in PBS was added for 5 min. The slides were rinsed twice and incubated with 10% normal goat serum (Thermo Fisher Scientific) for 15 min. The primary antibody rabbit anti-VE-cadherin (Abcam) was added. After 60-min incubation at RT and two washes, the secondary antibody Alexa Fluor 594 (anti-rabbit, Jackson) was added and incubated for 25 min at RT. 1% BSA and 0.05% Tween-20 were used for primary and secondary antibody dilution.

The chamber slide was rinsed three times and dried on room air for 15–30 min. The chambers were removed according to the manufacturer's instructions. VECTASHIELD Antifade Mounting Medium with DAPI (Vector Laboratories, Maravai LifeSciences) was added, and the slip was covered with Menzel 1-mm Microscope Coverslip (Thermo Fisher Scientific). Images were acquired with Zeiss LSM780 Laser Scanning Confocal Microscope (Carl Zeiss AG) with a 63x oil immersion objective using diode laser 405 nm for DAPI, argon laser 488 nm for FAM, and HeNe laser 594 nm for Alexa Fluor 594.

## Peptide homing

BALB/cJRj and diabetic male BKS(D)-Lepr[db]/JOrlRj mice with 7-d-old skin wounds were used for the peptide homing experiment. FAM-labeled peptides were dissolved in the PBS just before the experiment. 5 mg/kg (BALB/cJRj) or 1,5 mg/kg (BKS(D)-Lepr[db]/JOrlRj) of each peptide (CAR, CDD, CPK, CYH, and a control peptide) in PBS or plain PBS (control) was injected on the tail vein under sevoflurane anesthesia. 150 min later, the body was perfused as after in vivo microdialysis and the wounds, normal skin from the upper back, heart, left lung, left kidney, spleen, and a piece of liver were gathered in 4% PFA in PBS.

Normal or OIR C57BL/6Rj mice were injected intraperitoneally with peptide solution (5 mg/kg) at P17 (Vähätupa et al, 2021). Two hours after injection, the mice were perfused with DMEM containing 1% BSA while under deep anesthesia. The eyeballs were dissected and placed in 4% PFA (Vähätupa et al, 2021).

## Quantitative analysis of peptide homing

After 24-h (or 4 h for eyeballs) fixation in 4% PFA, the tissues were rinsed in 70% EtOH and embedded in paraffin. To avoid autofluorescence, the peptides were detected by staining the sections with rabbit anti-FITC/-FAM (71-1900; Thermo Fisher Scientific, 1:400 BALB/cJRj and C57BL/6Rj or 1:100 BKS(D)-Lepr[db]/JOrlRj) in Dako antibody diluent (Agilent) and Biocare Medical Rabbit on Rodent (RMR622H; Biocare Medical) with XM-Factor (XMF963C; Biocare Medical) as described previously (Robinson et al, 2010; Urakami et al, 2011; Toba et al, 2014; Pemmari, et al, 2020b). DAB (K3468; Agilent) was applied, and the slides were counterstained with hematoxylin (1.09253; Merck) and mounted. For CD31 stainings, rat anti-CD31 (550274; BD Pharmingen) in Dako antibody diluent and Histofine Simple Stain Mouse MAX PO (Nichirei) were used. DAB (K3468; Agilent) was applied, and the slides were counterstained with hematoxylin and mounted.

The brightfield images were acquired with NanoZoomer S60 (Hamamatsu Photonics). Slides were viewed and analyzed using QuPath software version 0.2.0 or later (https://qupath.github.io, Centre for Cancer Research and Cell Biology, Queen's University Belfast, Belfast, Northern Ireland, UK [Bankhead et al, 2017]). Analyzed areas were marked from high-magnification images using QuPath. Analysis algorithms were used, and the area of positive staining was determined as described previously (Vähätupa et al, 2021; Salomaa et al, 2022).

## Statistical analysis

For comparisons of multiple groups, statistical analysis was performed by two-way analysis of variance (ANOVA) complemented by the Bonferroni post hoc test for pairwise comparisons between the test groups. The possible difference in the homing of the different peptides was assessed using the log-transformed variables. $P$-values of less than 0.05 were considered statistically significant for all tests. The significance level shown refers to a two-tailed test.

# Data Availability

All data are available in the main text or the supplementary dataset. This study includes no data deposited in external repositories.

# Supplementary Information

# Acknowledgements

The authors thank professor Erkki Ruoslahti (Sanford Burnham Prebys Medical Discovery Institute, La Jolla, CA, USA) for his insightful comments on the article, Dr. Venkata Ramana Kotamraju (Sanford Burnham Prebys Medical Discovery Institute, La Jolla, CA, USA) for peptide synthesis, Anni Laitinen, Sari

Toivola, and Marianne Karlsberg for practical support and performing the histochemical work and immunohistological staining, and the Tampere Imaging Facility, BioCenter Finland, and FIMM Sequencing Service for offering their services. Mr. Peter Bocklund (CMA Microdialysis AB) is thanked for practical introduction to in vivo microdialysis and for providing the images for Fig S1. The work was funded by the Research Council of Finland (grant 287907 [to TAH Järvinen]), Competitive State Research Financing of the Expert Responsibility Area of Tampere University Hospital (grants 9R025, 9V010, 9X011, 9V010, T62774, and T63764 [to TAH Järvinen]), the Finnish Cultural Foundation, the Juhani Aho Foundation for Medical Research, the Finnish Medical Foundation, the Instrumentarium Science Foundation, the Paulo Foundation, the Emil Aaltonen Foundation, Diabetes Wellness Foundation, Päivikki and Sakari Sohlberg Foundation, Sigrid Juselius Foundation, Finnish Eye Foundation, Finnish Diabetic Research Foundation, and Tampere Tuberculosis Foundation. T Teesalu was funded by the Estonian Research Council (grants PRG230 and PRG1788), EuronanomedIII project ECM-CART, TRANSCAN3 project ReachGLIO (both coordinated by Estonian Research Council), and Bill and Melinda Gates Foundation (INV-026740).

## Author Contributions

T Pemmari: data curation, formal analysis, funding acquisition, validation, investigation, visualization, methodology, and writing—original draft, review, and editing.

S Prince: conceptualization, data curation, formal analysis, supervision, validation, investigation, methodology, and writing—review and editing.

N Wiss: investigation, visualization, and writing—review and editing.

K Kõiv: resources, investigation, methodology, and writing—review and editing.

U May: conceptualization, data curation, formal analysis, validation, investigation, and writing—review and editing.

T Mölder: resources, investigation, methodology, and writing—review and editing.

A Sudakov: resources, investigation, methodology, and writing—review and editing.

F Munoz Caro: investigation, visualization, and writing—review and editing.

S Lehtonen: investigation and writing—review and editing.

H Uusitalo-Järvinen: resources, data curation, formal analysis, supervision, funding acquisition, investigation, methodology, project administration, and writing—review and editing.

T Teesalu: conceptualization, resources, data curation, formal analysis, supervision, funding acquisition, validation, investigation, visualization, methodology, project administration, and writing—review and editing.

TAH Järvinen: conceptualization, resources, data curation, formal analysis, supervision, funding acquisition, investigation, visualization, methodology, project administration, and writing—original draft, review, and editing.

## Conflict of Interest Statement

The authors declare that they have no conflict of interest.

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
