## [Reviewer comments · Life Science Alliance]

Life Science Alliance

Screening of homing and tissue-penetrating peptides by microdialysis and in vivo phage display

Toini Pemmari, Stuart Prince, Niklas Wiss, Kuldar Koiv, Ulrike May, Tarmo Mölder, Aleksander Sudakov, Fernanda Munoz Caro, Soili Lehtonen, Hannele Uusitalo-Järvinen, Tambet Teesalu, and Tero Järvinen

DOI: <https://doi.org/10.26508/lsa.202201490>

Corresponding author(s): Tero Järvinen, Tampere University

Review Timeline:

Submission Date:	2022-04-20
Editorial Decision:	2022-07-06
Revision Received:	2025-01-13
Editorial Decision:	2025-01-14
Revision Received:	2025-01-31
Accepted:	2025-02-03

Transaction Report:

July 6, 2022

Re: Life Science Alliance manuscript #LSA-2022-01490-T

Tero Järvinen
Tampere University
Arvo Ylpön katu 31
Tampere, Pirkanmaa 33520
Finland

Dear Dr. Järvinen,

Thank you for submitting your manuscript entitled "Screening of homing and tissue-penetrating peptides by microdialysis and in vivo phage display" to Life Science Alliance. The manuscript was assessed by expert reviewers, whose comments are appended to this letter. We invite you to submit a revised manuscript addressing the Reviewer comments.

Thank you for this interesting contribution to Life Science Alliance. We are looking forward to receiving your revised manuscript.

Sincerely,

B. MANUSCRIPT ORGANIZATION AND FORMATTING:

Reviewer #1 (Comments to the Authors (Required)):

Pemmari et al incorporated microdialysis to the existing in vivo phage display method. This is a very creative and logical strategy for efficiently screening vascular targeting peptides that penetrate the blood vessel wall into tissue parenchyma. There are many peptides capable of homing to specific vascular beds; however, these peptides are not useful for drug delivery unless they can penetrate deeply into the tissue. Therefore, the idea for the peptide screening method reported in this manuscript is of great interest. For instance, this method could be used to discover blood-brain-barrier penetrating peptides if used in the brain, in theory.

However, this study has some key weaknesses that need to be addressed. Some of the claims by the authors are not sufficiently supported due to the absence of control peptides and quantitative analyses.

Major points

Fig. 1
Cultured endothelial cells could uptake any peptide added to the culture. Therefore, 3 or more other peptides should be included in the study for comparison. The peptides that didn't show up in the microdialysate will be good controls. Also, representative fluorescence cell images alone are insufficient for this study. The analysis should be carried out by determining fluorescence intensity per cell in triplicated wells for quantitative comparisons and statistical significance. It may be necessary to perform this study in time-course to see clear differences in the peptide internalization efficiencies.

The figure labels say "FITC". Should this be FAM?

Fig. 2
Similar to Fig. 1, this study should include a few other peptides to compare with CDD, CPK, and CYH, and the peptide accumulation should be quantified per granulation tissue area by image analyses to demonstrate the efficient homing and tissue penetration of the three peptides compared with the non-microdialysate peptides.

Fig. EV2
The authors claim that the three peptides localize in the region of granulation tissue rich in angiogenic neovessels. However, they provide no images of the regions without abundant angiogenesis in comparison to support the claim.

The authors claim and their in vivo analyses suggest that the three peptides (CDD, CPK, and CYH) efficiently extravasate through the vessel wall after homing to the target endothelium, but this activity of the peptides was not tested in vitro. To strengthen the main conclusion, this should be tested by penetration of FAM-peptides across the confluent endothelial monolayer using Transwell culture inserts. A few other peptides should be included to compare the extravasation activities of these peptides.

Fig. EV3
The authors wrote, "For all tested peptides, we observed efficient peptide accumulation in the granulation tissue..." However, only CDD was efficient in diabetic skin wounds according to Fig. EV3. Again, quantitative measurements of peptide accumulation should be carried out to accurately evaluate the homing/extravasation activities of the peptides. It appears that the staining of CPK shown here might be non-specific. Do you find the same staining pattern of this peptide throughout the granulation tissue? It is difficult to judge from a single picture. A quantitative analysis of the entire scanned images of several wound tissue samples will be helpful.

Minor comments

Fig. EV1

For the readers, it would be helpful to include a picture of the dialysis probe before inserting it into the skin.

Fig. 3

The description of Fig. 3a is missing from the figure legend.

Little information is provided for the number of animals used, and it is not clear whether each experiment was repeated and the average value of multiple experiments was shown or a representative result was shown.

Reviewer #2 (Comments to the Authors (Required)):

In this report, the authors incorporate an interesting approach to assess peptides that penetrate tissue parenchyma, which can be of utility for in vivo phage display screenings designed to target normal tissues or certain pathologies where the vasculature or pathology-associated changes may represent a barrier to entering the tissue compartment. They propose to use in vivo phage display screenings in mice using a skin wound model combined with microdialysis to recover phage particles that localize to the perivascular region of the wound. The paper would benefit from additional discussion, experimental details, and potentially additional data related to the selection/recovery process and validation and specificity of the phage identified in the proposed setting.

It is not clearly demonstrated that the microdialysis isolates clones that home to tissue, because they seem to be recovered as "unbound" using this methodology. Thus, after crossing the vasculature, do the phage bind to a cell surface receptor in the parenchyma, or are released post transcytosis? The authors show cell surface binding for the three selected peptides in vitro in HUVEC. It may be worth exploring internalization of these peptides in epithelial or other stromal cell lines to inform any future therapeutic/diagnostic applications.

In moving forward, the authors must strongly consider additional experimental work to support mechanism, such as a transwell experiments with endothelial cells to confirm that the three discovered peptides undergo receptor-mediated transport post internalization.

The authors should also add some level of quantification for the homing of the three peptides to skin wound in comparison to the healthy tissues. It is very difficult to interpret Fig. 3 (in addition to the other figures) due to low resolution and magnification.

The confocal and IHC images are difficult to visualize; higher resolution and higher magnification images are needed to adequately visualize the phage particles.

The criteria for selecting the three peptides were based on motifs similarities. However, there is missing information about the frequency of these motifs in high throughput which seems likely to show multiple potential motifs. Data need to be provided to preclude any bias.

The microdialysis apparatus with the 3 MDa filter only captures 1 in 27 phage based on the authors' initial experiment mentioned in the manuscript. As such, it is fair to assume that 96-97% of phage in the PBS are not being captured? This is not necessarily a liability that takes away from the selection system, however, it would be helpful to articulate either way.

What are the advantages and disadvantages of this methodology in comparison to others, such as cell-sorting flow cytometry? Discussing this point would strengthen the manuscript. This reviewer would also suggest a more positive tone towards previous in vivo phage display work. The existing literature do not take away the merit of a new/different way of carrying out library selections for ligand-directed targeting and receptor mapping. Please revise the language.

If the library diversity is 3.9×10^8 , it would be helpful to clarify the rationale for the 3.9×10^{10} initial phage input. The Methods state that the library diversity was 1×10^9 . Can the authors clarify which one is correct?

cross-comment section:

I have a comment on the following Referee #1 statement ""There are many peptides capable of homing to specific vascular beds; however, these peptides are not useful for drug delivery unless they can penetrate deeply into the tissue".

This is simply not the case, and we will get a hold of representative literature, including clinical trials carried out in the US, European Union, and Japan, that use peptides identified by in vivo phage display, for ligand-directed drug delivery.

In addition, work by our laboratory has covered elegant mechanistic work related to the transcytosis of peptide ligands, for imaging, therapy, and vaccination strategies, in the context of brain and lung vascular targeting.

A point-by-point reply to concerns raised by the Life Science Alliance:

Reviewer #1 (Comments to the Authors (Required)):

Pemmari et al incorporated microdialysis to the existing in vivo phage display method. This is a very creative and logical strategy for efficiently screening vascular targeting peptides that penetrate the blood vessel wall into tissue parenchyma. There are many peptides capable of homing to specific vascular beds; however, these peptides are not useful for drug delivery unless they can penetrate deeply into the tissue. Therefore, the idea for the peptide screening method reported in this manuscript is of great interest. For instance, this method could be used to discover blood-brain-barrier penetrating peptides if used in the brain, in theory.

However, this study has some key weaknesses that need to be addressed. Some of the claims by the authors are not sufficiently supported due to the absence of control peptides and quantitative analyses.

Reply: We thank the reviewer # 1 for her/his generally positive comments on our manuscript. Concerning the weaknesses, we feel that we have addressed them by a new set of experiments.

Major points

Fig. 1. Cultured endothelial cells could uptake any peptide added to the culture. Therefore, 3 or more other peptides should be included in the study for comparison. The peptides that didn't show up in the microdialysate will be good controls. Also, representative fluorescence cell images alone are insufficient for this study. The analysis should be carried out by determining fluorescence intensity per cell in triplicated wells for quantitative comparisons and statistical significance. It may be necessary to perform this study in time-course to see clear differences in the peptide internalization efficiencies.

Reply: *Due to a rather poor quality of the original figures demonstrating the in vitro cell penetration by the microdialysis-captured peptides, we have re-performed the in vitro cell penetration experiments by using both epithelial (CHO-K) and endothelial (HUVECs) cells to demonstrate the in vitro cell penetration by the microdialysis-captured novel peptides (please see Figs. 1 and EV3). As the purpose of the study was not to identify the best cell penetrating peptide, but to demonstrate that the microdialysis-captured peptides are cell penetrating peptides, no quantitative analysis of cell penetration was performed.*

The figure labels say "FITC". Should this be FAM?

Reply: *Yes. Revised accordingly.*

Fig. 2 Similar to Fig. 1, this study should include a few other peptides to compare with CDD, CPK, and CYH, and the peptide accumulation should be quantified per granulation tissue area by image analyses to demonstrate the efficient homing and tissue penetration of the three peptides compared with the non-microdialysate peptides.

Reply: *We understand the point made by reviewer # 1. We have performed quantitative analysis of peptide homing to skin wounds (Fig. 3). In addition to that, we have performed new experiments where the peptide homing was explored quantitatively to experimental retinopathy-model (Figs. 3 and 4).*

Fig. EV2 The authors claim that the three peptides localize in the region of granulation tissue rich in angiogenic neovessels. However, they provide no images of the regions without abundant angiogenesis in comparison to support the claim.

Reply: *We want to remind the reviewer that the name "granulation tissue" is derived from the granular appearance of the blood vessels filling the loose connective tissue forming at the regenerating skin wound.*

Thus, it is hard to find an area of granulation tissue without blood vessels as the granulation tissue is formed by blood vessels. It is true that the peptide homing is the strongest in the areas of granulation tissue rich of blood vessels, but we have decided to omit that statement.

The authors claim and their in vivo analyses suggest that the three peptides (CDD, CPK, and CYH) efficiently extravasate through the vessel wall after homing to the target endothelium, but this activity of the peptides was not tested in vitro. To strengthen the main conclusion, this should be tested by penetration of FAM-peptides across the confluent endothelial monolayer using Transwell culture inserts. A few other peptides should be included to compare the extravasation activities of these peptides.

Reply: *The fundamental problem with the transwell experiments is that they require confluent cell cultures. We want to remind that we have screened for peptides that home to angiogenic blood vessels in this study. Our experience with similar homing peptides that target angiogenic vasculature (Järvinen & Ruoslahti, *Am J Pathol*, 171:702-711, 2007 and Sugahara et al. *Science* 2010;328(5981):1031-1035, Teesalu T, et al. *Proc Natl Acad Sci U S A*. 2009;106(38):16157-16162, refs. ^{8,9,22}) is that the expression of their respective receptors is lost from the cultured cells when they reach confluence in the culture (i.e. the peptide binding disappears). To address the valid concern raised by the reviewer, we re-performed in vitro cell penetration assays in HUVECs (Fig. EV3). In addition to that, we performed the same experiment in epithelial cells (Fig. 1). Finally, we addressed whether the peptides penetrate cells/tissues with an additional hindrance in the blood vessels, i.e. blood-retina barrier, by performing homing experiments in the retinopathy-model (Fig. 3 & 4). We believe that by performing these experiments we have adequately established both the homing and the tissue-penetrating capabilities of the microdialysis-captured peptides.*

Fig. EV3

The authors wrote, "For all tested peptides, we observed efficient peptide accumulation in the granulation tissue..." However, only CDD was efficient in diabetic skin wounds according to Fig. EV3. Again, quantitative measurements of peptide accumulation should be carried out to accurately evaluate the homing/extravasation activities of the peptides. It appears that the staining of CPK shown here might be non-specific. Do you find the same staining pattern of this peptide throughout the granulation tissue? It is difficult to judge from a single picture. A quantitative analysis of the entire scanned images of several wound tissue samples will be helpful.

Reply: *We performed quantitative analysis of peptide homing in normal skin excision wound model (Fig. 3), but not in the diabetic wound model. The reason for that is that the diabetic wound model is very complicating one, i.e. not very reproducible. There is a massive fat infiltration in the diabetic wound model we have employed. This makes the quantitative analysis very unreliable compared to the quantitative analysis while employing the standard excision skin wound model, where the granulation tissue formation/angiogenesis is reproducible.*

Minor comments

Fig. EV1

For the readers, it would be helpful to include a picture of the dialysis probe before inserting it into the skin.

Reply: *We have provided pictures of microdialysis-probe as well as the principle of the microdialysis for the reader, please see the new Fig. EV1.*

Fig. 3

The description of Fig. 3a is missing from the figure legend.

Reply: *We thank the reviewer # 1 for pointing out this. We have revised the manuscript accordingly.*

Little information is provided for the number of animals used, and it is not clear whether each experiment was repeated and the average value of multiple experiments was shown or a representative result was shown.

Reply: *As we have carried out quantitative analysis of homing, we have indicated the number of experiments and the number of the animals included in the experiments, please see Figs. 3-4.*

Reviewer #2 (Comments to the Authors (Required)):

In this report, the authors incorporate an interesting approach to assess peptides that penetrate tissue parenchyma, which can be of utility for in vivo phage display screenings designed to target normal tissues or certain pathologies where the vasculature or pathology-associated changes may represent a barrier to entering the tissue compartment. They propose to use in vivo phage display screenings in mice using a skin wound model combined with microdialysis to recover phage particles that localize to the perivascular region of the wound. The paper would benefit from additional discussion, experimental details, and potentially additional data related to the selection/recovery process and validation and specificity of the phage identified in the proposed setting.

Reply: *We thank the reviewer # 2 for her/his generally positive comments on our manuscript.*

It is not clearly demonstrated that the microdialysis isolates clones that home to tissue, because they seem to be recovered as "unbound" using this methodology. Thus, after crossing the vasculature, do the phage bind to a cell surface receptor in the parenchyma, or are released post transcytosis? The authors show cell surface binding for the three selected peptides in vitro in HUVEC. It may be worth exploring internalization of these peptides in epithelial or other stromal cell lines to inform any future therapeutic/diagnostic applications.

Reply: *First of all, we re-performed the cell penetration experiment in the HUVECs, please see Fig EV3. Then we used epithelial cell line, CHO-Ks, to demonstrate the cell penetration to the epithelial cells by the microdialysis-identified homing and cell penetrating peptides, please see Fig. 1.*

In moving forward, the authors must strongly consider additional experimental work to support mechanism, such as a transwell experiments with endothelial cells to confirm that the three discovered peptides undergo receptor-mediated transport post internalization.

Reply: *As already mentioned above, we extended the in vitro experiments to epithelial cells (Fig. 1), then we quantified the homing in skin wound (Fig. 3). Finally, we addressed whether these novel peptides penetrate blood-retina barrier in the experimental retinopathy-model (Figs. 3 & 4). We quantified the homing and the peptide accumulation in the retinopathy-model to address the concerns raised by the reviewer # 2.*

Concerning the transwell-experiments proposed by the reviewer # 2, we kindly ask to look at the thorough response we have provided to the reviewer # 1.

The authors should also add some level of quantification for the homing of the three peptides to skin wound in comparison to the healthy tissues. It is very difficult to interpret Fig. 3 (in addition to the other figures) due to low resolution and magnification.

Reply: *We have quantified the homing not just in the skin wound but also in the retinopathy-model, please see Fig. 3 in the revised version of the manuscript.*

The confocal and IHC images are difficult to visualize; higher resolution and higher magnification images are needed to adequately visualize the phage particles.

Reply: *We agreed with the assessment of the reviewer # 2 presented above. Thus, we re-performed the in vitro cell penetration experiments in the HUVECs (Fig. EV3), did similar experiments in the CHO-K cells (Fig. 1) and did additional homing experiments in the retinopathy-model (Figs. 3 & 4). We also believe that the quality of the presented images has improved substantially from the original version of the manuscript.*

The criteria for selecting the three peptides were based on motifs similarities. However, there is missing information about the frequency of these motifs in high throughput which seems likely to show multiple potential motifs. Data need to be provided to preclude any bias.

Reply: *The requested data is provided in the Table 3.*

The microdialysis apparatus with the 3 MDa filter only captures 1 in 27 phage based on the authors' initial experiment mentioned in the manuscript. As such, it is fair to assume that 96-97% of phage in the PBS are not being captured? This is not necessarily a liability that takes away from the selection system, however, it would be helpful to articulate either way.

Reply: *The general principle of the microdialysis is that the microdialysis "capture rate" is always < 40 % of the actual extracellular concentration of the analyte. As we were rescuing such large entity as phage, it was expected that only a small fraction of phages would be captured by the microdialysis probe.*

What are the advantages and disadvantages of this methodology in comparison to others, such as cell-sorting flow cytometry? Discussing this point would strengthen the manuscript. This reviewer would also

suggest a more positive tone towards previous in vivo phage display work. The existing literature do not take away the merit of a new/different way of carrying out library selections for ligand-directed targeting and receptor mapping. Please revise the language.

Reply: *We are hesitant to compare the new technology presented in the manuscript to conventional in vivo phage display due to preliminary status of our technology and the fact that we did not perform comparative studies between different screening methods. However, we do state: “.....we developed an approach for identification of target-selective homing peptides based solely on in vivo display high throughput sequencing (HTS) data. However, even the introduction of HTS for the analysis of in vivo display data could not shorten the need for multiple rounds of screening to identify homing selectivity”²¹.*

If the library diversity is 3.9×10^8 , it would be helpful to clarify the rationale for the 3.9×10^{10} initial phage input. The Methods state that the library diversity was 1×10^9 . Can the authors clarify which one is correct?

Reply: *First of all, all peptide libraries used in the manuscript are presented with a correct diversity. We want to emphasize that we used three different phage libraries in the manuscript: 1) Phage library expressing random peptide sequences for in vitro work, 2) custom-made library containing phage clones expressing known homing peptide sequences and 3) custom-made CX7^{trinuc}C peptide T7 phage library, in which random amino acids are encoded by equally represented trinucleotides to prevent codon bias. Furthermore, this library was cloned into the genome of the T7 bacteriophage engineered for longer circulation by introducing a point mutation in the phage coat protein. The different peptide libraries are explained on page 18 of the revised version of the manuscript.*

cross-comment section:

I have a comment on the following Referee #1 statement ""There are many peptides capable of homing to specific vascular beds; however, these peptides are not useful for drug delivery unless they can penetrate deeply into the tissue".

This is simply not the case, and we will get a hold of representative literature, including clinical trials carried out in the US, European Union, and Japan, that use peptides identified by in vivo phage display, for ligand-directed drug delivery.

In addition, work by our laboratory has covered elegant mechanistic work related to the transcytosis of peptide ligands, for imaging, therapy, and vaccination strategies, in the context of brain and lung vascular targeting.

Reply: No action taken, as the points raised above by the reviewer # 2 relate to the comment made by the reviewer # 1.

January 14, 2025

RE: Life Science Alliance Manuscript #LSA-2022-01490-TR

Prof. Tero A Järvinen
Tampere University
Arvo Ylpön katu 31
Tampere, Pirkanmaa 33520
Finland

Dear Dr. Järvinen,

Thank you for submitting your revised manuscript entitled "Screening of homing and tissue-penetrating peptides by microdialysis and in vivo phage display". We would be happy to publish your paper in Life Science Alliance pending final revisions necessary to meet our formatting guidelines.

- please be sure that the authorship listing and order is correct
- please upload all figure files as individual ones, including the supplementary figure files; all figure legends should only appear in the main manuscript file
- LSA allows supplementary figures, but not EV Figures; please update your callouts for the Supplementary Figures in the manuscript Fig EV1A = Fig S1A)
- please add the Twitter and Bluesky handles of your host institute/organization as well as your own or/and one of the authors in our system
- please be sure that the authorship listing and order is correct
- please add your main, supplementary figure, and table legends to the main manuscript text after the references section and remove them from the manuscript file
- please consult our manuscript preparation guidelines <https://www.life-science-alliance.org/manuscript-prep> and make sure your manuscript sections are in the correct order
- please add a callout for Figure 3A-D; 4A-D; S1A-C and S5A-B to your main manuscript text
- please incorporate the supplementary references into the main Reference list

FIGURE CHECK:

- please double-check that the boxes in Figure S6A match neatly where the magnified images in panel B were taken from

A. FINAL FILES:

-- Summary blurb (enter in submission system): A short text summarizing in a single sentence the study (max. 200 characters including spaces). This text is used in conjunction with the titles of papers, hence should be informative and complementary to

the title. It should describe the context and significance of the findings for a general readership; it should be written in the present tense and refer to the work in the third person. Author names should not be mentioned.

B. MANUSCRIPT ORGANIZATION AND FORMATTING:

Sincerely,

February 3, 2025

RE: Life Science Alliance Manuscript #LSA-2022-01490-TRR

Prof. Tero A Järvinen
Tampere University
Faculty of Medicine and Health Technology
Arvo Ylpön katu 31
Tampere, Pirkanmaa 33520
Finland

Dear Dr. Järvinen,

Thank you for submitting your Methods entitled "Screening of homing and tissue-penetrating peptides by microdialysis and in vivo phage display". It is a pleasure to let you know that your manuscript is now accepted for publication in Life Science Alliance. Congratulations on this interesting work.

DISTRIBUTION OF MATERIALS:

Again, congratulations on a very nice paper. I hope you found the review process to be constructive and are pleased with how the manuscript was handled editorially. We look forward to future exciting submissions from your lab.

Sincerely,
